# Dual-Channel Stretchable, Self-Tuning, Liquid Metal Coils and Their Fabrication Techniques

**DOI:** 10.3390/s23177588

**Published:** 2023-09-01

**Authors:** Elizaveta Motovilova, Terry Ching, Jana Vincent, James Shin, Ek Tsoon Tan, Victor Taracila, Fraser Robb, Michinao Hashimoto, Darryl B. Sneag, Simone Angela Winkler

**Affiliations:** 1Department of Radiology, Weill Cornell Medicine, New York, NY 10065, USA; 2Department of Radiology and Imaging, Hospital for Special Surgery, New York, NY 10021, USA; 3Pillar of Engineering Product Development, Singapore University of Technology and Design, Singapore 487372, Singapore; 4Digital Manufacturing and Design (DManD) Centre, Singapore University of Technology and Design, Singapore 487372, Singapore; 5Department of Biomedical Engineering, National University of Singapore, Singapore 117583, Singapore; 6GE Healthcare, Aurora, OH 44202, USA

**Keywords:** radiofrequency coil, liquid metal, stretchable electronics, self-tuning, 3D printing, direct ink writing

## Abstract

Flexible and stretchable radiofrequency coils for magnetic resonance imaging represent an emerging and rapidly growing field. The main advantage of such coil designs is their conformal nature, enabling a closer anatomical fit, patient comfort, and freedom of movement. Previously, we demonstrated a proof-of-concept single element stretchable coil design with a self-tuning smart geometry. In this work, we evaluate the feasibility of scaling this coil concept to a multi-element coil array and the associated engineering and manufacturing challenges. To this goal, we study a dual-channel coil array using full-wave simulations, bench testing, in vitro, and in vivo imaging in a 3 T scanner. We use three fabrication techniques to manufacture dual-channel receive coil arrays: (1) single-layer casting, (2) double-layer casting, and (3) direct-ink-writing. All fabricated arrays perform equally well on the bench and produce similar sensitivity maps. The direct-ink-writing method is found to be the most advantageous fabrication technique for fabrication speed, accuracy, repeatability, and total coil array thickness (0.6 mm). Bench tests show excellent frequency stability of 128 ± 0.6 MHz (0% to 30% stretch). Compared to a commercial knee coil array, the stretchable coil array is more conformal to anatomy and provides 50% improved signal-to-noise ratio in the region of interest.

## 1. Introduction

Radio frequency (RF) coils play a significant role in any magnetic resonance imaging (MRI) system as they are used to excite nuclear magnetic moments (transmit coils) and to receive MR signals (receive coils) [1]. Receive coil size is determined by the targeted field of view (FOV) and desired penetration depth. To achieve optimal signal-to-noise ratio (SNR), the coil diameter *D* is roughly equal to the penetration depth *p* as follows $D\approx 2p/\sqrt{5}$ [2]. Coil proximity to the targeted FOV and high coil filling factor results in high receive sensitivity and performance stability under varying loading conditions between different subjects. Modern clinical RF receive coils and coil arrays are encased within (semi-)rigid protective housings that are roughly form-fitted to a specific anatomical area, e.g., head, breast, wrist, knee [3], and are designed to fit the average population. However, coil proximity is not always optimal due to variations in subjects’ weight and size, which routinely results in reduced SNR. Moreover, rigid coil housing causes patient discomfort and limits the ability to perform dynamic imaging of musculoskeletal (MSK) joint soft tissues in motion.

Recent advances in the design and development of RF coils have been driven by the desire to improve patient experience and have resulted in a number of flexible and stretchable solutions [4]. The degree of flexibility depends on the design approach and the choice of coil materials. In [5], a thin and flexible coil was printed on both sides of a Kapton substrate such that to create the required in line capacitors for coil resonance. The screen-printed coil fabrication method developed by Corea et al. [6,7] avoids bulky copper wires, porcelain capacitors, and thick substrates, allowing for more flexibility and closer coil placement, which is especially suitable for pediatric imaging [8]. Several semi-rigid coil arrays have been proposed that can be wrapped around a 2D curved surface [9,10,11,12,13,14,15] or mechanically adjusted to better fit the targeted areas [16,17,18,19], thereby enabling accommodation of different patient sizes. High-impedance coil (HIC) technology using coaxial cables offers improved inter-element isolation and high flexibility while demonstrating the feasibility of dynamic imaging applications [20,21,22,23,24]. A commercially available flexible coil design called adaptive image receive (AIR) coil technology (General Electric (GE) Healthcare) is ultra-lightweight, requires no lumped components on the coil conductor, and easily conforms to various patient anatomies like a blanket [25]. However, none of these methods represent a truly stretchable coil design. research has shown early stretchable coil conductor prototypes using unconventional materials such as copper braids [26], conductive thread [27,28,29,30], liquid metal [31,32,33,34,35,36], and conductive elastomer [37]. Even though such alternative materials are lossier than traditionally used copper conductors, it was shown that they can offer similar SNR and provide additional advantages such as light weight and radiological transparency [38]. The biggest roadblock that stretchable coils suffer from is a resonance frequency shift as the length of the conductor and thus the inductance of the coil is increasing under stretch. Several solutions were proposed to mitigate frequency shift, including implementing field programmable gate array (FPGA)-based tuning/matching circuits [39] and π-matching networks [16]. 

Recently, we proposed a stretchable coil element design with a smart self-tuning geometry that maintains a stable resonance frequency for degrees of elongation up to 30% [32]. Modern MRI receive coils are built from multiple coil elements to maximize signal-to-noise ratio (SNR). This work expands on the previously published single-element concept to establish feasibility for a multi-element array coil and focuses on the fabrication challenges encountered in combining the elements with optimal overlap and mechanical stability. To this goal, we design a dual-channel coil array using numerical simulations and different fabrication techniques. In our previous work with the single coil element, we used a polymer casting method to create the microchannel structure housing the conducting liquid metal. However, multi-channel coil arrays require overlapping conductors that are not connected electrically, which cannot be produced with the mold casting method. Therefore, we explore three alternative fabrication techniques and construct three corresponding dual-channel coil array prototypes: (1) single layer casting (SLC) with jumper wires, (2) double layer casting (DLC), and (3) direct-ink-writing (DIW). We demonstrate the performance of all three coil array prototypes on the bench, as well as in vitro and in vivo using a 3 tesla (T) MRI system.

## 2. Materials and Methods

### 2.1. Simulations

Full wave numerical simulations were performed using COMSOL 6.0 Electromagnetic Waves Frequency Domain (EMW) Physics module (Burlington, MA, USA). Figure 1 shows (a) 3D view of the simulation model, (b) top view of the dual-channel coil, and (c) enlarged view of the interdigital capacitor. The coil geometry consisted of a 7 cm × 6 cm rectangular loop with an integrated interdigital capacitor as outlined in [32]. The conducting traces were realized as 0.5 mm wide perfect electric conductor (PEC) microchannels embedded in a 3 mm thick polymer substrate ($\varepsilon_{r}=3.4$). The coil array was positioned on a homogenous phantom (width *W* = 22 cm, length *L* = 33 cm, height *H* = 16 cm, $\varepsilon_{r}=78$, σ=0.46 S/m). Two lumped ports were used for coil excitation with 1 W input power and a characteristic impedance of 50 Ω. Two lumped elements functioning as tuning and matching capacitors were connected between the coil and the port of each coil element. An air-filled sphere with a radius of 40 cm and 8 cm thick perfectly matched layer (PML) boundary conditions was used to surround the simulation model (not shown in Figure 1a for simplicity). A user defined frequency sweep ranging from 118 MHz to 138 MHz with increments of 0.1 MHz was simulated to study the change in the scattering parameters, *S_ij_*, when stretched. In this proof-of-concept study, we focus only on a unidirectional stretching in *x*-direction. The stretching mechanism was implemented by parameterizing all geometry elements with dimensionless variables $\lambda_{x},\lambda_{y},\lambda_{z},$ where the subscripts correspond to the three Cartesian directions *x, y,* and *z*, respectively. For a uniaxial stretch along the *x*-direction, the variable $\lambda_{x}$ was swept from 1 to 1.5, representing elongations from 0% to 50%. Assuming the materials are elastically homogenous and virtually incompressible, we impose the constraint $\lambda_{x}\cdot \lambda_{y}\cdot \lambda_{z}=1$, from where it follows that *y* and *z* dimensions will change (shrink) according to $\lambda_{y}=\lambda_{z}=1/\sqrt{\lambda_{x}}$. Figure 1d–f shows representative coil stretching configurations, i.e., degrees of (d) 10%, (e) 30%, and (f) 50% stretch. Coil elements were decoupled using the critical overlap decoupling technique [40] with the critical overlap distance determined from *S*_21_ simulations. The overlap distance was varied from 8 mm to 17 mm, and the critical overlap distance was determined from the minimum of *S*_21_. This critically overlapped dual coil array was then linearly stretched in *x*-direction from 0% to 50% and *S*-parameters as well as resonance frequency changes were recorded. Individual and combined coil sensitivity profiles (*B*_1_-field maps) were calculated. We also studied whether the variation in layer thickness affects coil decoupling and resonance shift.

### 2.2. Fabrication Techniques

In our prior research involving the single coil element [32], we employed the polymer casting method to generate the microchannel structure. However, employing the same fabrication technique does not allow for the creation of overlapping microchannels, which are essential for a dual-channel coil array. As a result, we delve into investigating three different fabrication techniques and construct three corresponding prototypes of dual-channel coil arrays. Three different fabrication techniques are (1) single-layer casting (SLC) with a jumper wire, (2) double-layer casting (DLC), and (3) direct-ink-writing (DIW). The first two techniques, SLC and DLC, use the same casting/molding fabrication principle as described in reference [32], where soft lithography strategies were used [41]. The DIW method uses the manufacturing principle described in [42]. All three techniques are described in detail below.

#### 2.2.1. Single Layer Casting

In the SLC method, the conducting traces are located on the same layer. In the region where the coil elements overlap, a gap is introduced in the liquid metal conductor to allow insertion of a jumper wire. Figure 2 shows a step-by-step illustration of the SLC fabrication method. First, two customized molds were designed with SolidWorks (Dassault Systems, Vélizy-Villacoublay, France): (1) an upper mold containing the negative of the conductive traces and (2) a lower mold without any features. The molds were fabricated using a high-resolution 3D printer (Prusa i3 MK3S, using a 25 µm nozzle and a 50 µm layer height) with polylactic acid (PLA) material (Prusa Polymers, Prague, Czech Republic). We used Ecoflex 00-30 (Smooth-ON, Macungie, PA, USA) material as the stretchable silicone matrix because it is biocompatible, can be stretched up to 900%, and does not require any specialist equipment to mix and prepare besides a table-top vacuum chamber. The two parts (A and B) of the Ecoflex 00-30 silicone were combined at a A:B = 1:1 ratio and thoroughly mixed by hand for 3 min. After degassing the mixture in a vacuum chamber for 10 min, it was poured into the prepared upper mold, as shown in Figure 2a. After the silicone was fully cured, it was peeled off the upper mold, as shown in Figure 2b. Another Ecoflex 00-30 mixture was prepared similarly and poured into the lower mold, followed by a partial curing for a duration of 1 h at room temperature. To achieve a robust interface between the upper and lower layers, the fully cured upper layer was attached to the partially cured lower layer, as shown in Figure 2c, and the assembly was then cured at room temperature for another 3 h. To establish the necessary connection between the traces of the same coil element at the overlap area, a thin and flexible silicone tube (inner diameter *ID* = 0.5 mm, length *L* = 5 mm) was used, thus creating a “jumper wire”, as shown in Figure 2d. More illustrative photographs of this process are demonstrated in the following Section 3.2.1. Liquid metal (GaIn) was injected into the channels using a syringe with a G25 needle, as shown in Figure 2e. Figure 2f illustrates bonding of the two layers to form a single-layer coil.

#### 2.2.2. Double Layer Casting

In the DLC method, two separate single-element loops are fabricated and placed on top of one another thus creating a double-layer structure. Figure 3 shows a step-by-step illustration of the DLC fabrication method. First, two customized molds were designed with SolidWorks (Dassault Systems): (1) an upper mold containing the negative of the conducting traces and (2) a lower mold without any features. The molds were fabricated using the same 3D printer and settings as in the previous SLC method. To facilitate coil alignment for critical overlap decoupling and to maintain the same substrate thickness throughout the array, the two layers were matched in size, with one layer containing the coil conductors on its right and the other containing its conducting traces on the left side. Similar to the SLC method, the Ecoflex 00-30 mixture was prepared, degassed, and poured into the upper mold a shown in Figure 3a. After being fully cured and peeled off (Figure 3b) the upper layer was placed on top of the partially cured lower Ecoflex layer (Figure 3c). Liquid metal (GaIn) was injected using a syringe with a needle, Figure 3d. To make the second coil, steps (a)–(d) were repeated. The two coils were then aligned and stacked (Figure 3e) to create a dual-channel array with optimized overlap decoupling (see Section 3.1 on how this overlap distance was obtained.

#### 2.2.3. Direct Ink Writing

In the third method, very thin coil elements were fabricated using a direct ink writing (DIW) technique [42]. Figure 4 shows a step-by-step schematic of the DIW fabrication method. DragonSkin^TM^ 30 (Smooth-On, Macungie, PA, USA) silicone material was chosen for its stretchability that can surpass 300% and a higher tensile strength (500 psi) compared to Ecoflex (200 psi). To create the base silicone layer, DragonSkin^TM^ was spin-coated on a glass panel at 700 rpm for 40 s, as illustrated in Figure 4a–b, and left to fully cure. The silicone walls of the microchannels, ultimately housing the liquid metal conductors, were printed using a DIW printer (SHOTmini200ΩX, Musashi Engineering Inc., Tokyo, Japan) as illustrated in Figure 4d. A biocompatible, rapidly curing silicone resin (SpeedSeal) was employed to form these microchannel walls. This resin exhibits appropriate physical and chemical characteristics, guaranteeing: (1) minimal spreading upon printing, (2) strong adhesion to other silicone-based substrates, and (3) stretchability after curing. While this SpeedSeal microchannel layer was still uncured, an additional top layer comprised a thin silicone sheet (DragonSkin) was affixed to seal the channels. This process (a)–(c) was repeated to create the second coil element. Liquid metal (GaIn) was injected using a syringe with a needle (Figure 4c). The two elements were overlapped to achieve critical overlap decoupling, similarly to the process shown in Figure 3e, and securely attached using SealPoxy (SmoothON, Macungie, PA, USA).

#### 2.2.4. Conductors and Circuitry

In all coil prototypes, the microchannels were filled with GaIn liquid metal. Gallium-based liquid metal alloys such as eGaIn and Galinstan have achieved broad usage in flexible and stretchable electronics owing to their exceptional deformability, high electrical conductivity, minimal vapor pressure, low viscosity, and low toxicity [43]. The alloy used in this work consists of 75.5% Gallium and 24.5% Indium by weight. GaIn has a low viscosity of 1.99 × 10^−3^ Pa·s [44], which allows for easy flow through both the needle and the microchannels. Copper wires were inserted at the terminals. GaIn does not cause corrosion or oxidation of the copper wires [43]. To prevent any potential leakage of the liquid metal and ensure the stability of the wires, we sealed all channel openings and wire connections using silicone epoxy (SilPoxy^TM^, SmoothON). Subsequently, the copper wires were interconnected to a printed circuit board that incorporated circuitry for tuning, matching, detuning, and preamplifier circuitry.

### 2.3. Benchtop Measurements

The mechanical stability of all three prototypes was assessed through a stretch test. To accomplish this, a stretch test setup was employed, enabling gradual elongation of the coils in a single direction. The coils were connected to a vector network analyzer (Keysight E5071C), and the *S*-parameters were recorded throughout the stretching process. The stretch test involved elongating the coil from 0% to 30%, with the elongation (∆*x*) measured in millimeters and converted to a percentage of the initial coil length $x_{0}$, as $stretch\left(\%\right)=100\%\cdot \Delta x/x_{0}$. This experiment was performed for each of the three coil arrays. *S*_11_ provides the resonance frequency as well as tuning and matching performance for each element, while *S*_21_ yields decoupling performance between elements.

### 2.4. In Vitro Imaging

The coil arrays were positioned on a standard rectangular silicone phantom (width *W* = 22 cm, length *L* = 33 cm, height *H* = 16 cm) for imaging at 3 T (scanner model MR750, GE Healthcare, Chicago, IL, USA), as shown in Figure 5a. A 3D spoiled gradient echo sequence was used (time to repeat *TR* = 6.3 ms, echo time *TE* = 2.4 ms (in phase), field of view *FOV* = 20 cm, pixel size 0.8 mm × 0.8 mm, flip angle *FA* = 12 deg, bandwidth *BW* = 31.3 kHz, slice thickness = 1 mm, number of averages *NEX* = 1). For stretching tests, a plastic stretching rig was used, and the coil array was gradually stretched from 0 cm to 5 cm in increments of Δx = 1 cm. The SNR values were calculated as the ratio of the mean pixel value of signal within the specified region of interest (ROI) divided by the standard deviation of the noise calculated in the background region of the image, well removed from the phantom and any visible artifacts [45]. The SNR maps were then calculated by dividing the entire image by the standard deviation of the noise calculated as described above.

### 2.5. In Vivo Imaging

The in vivo study was performed with ethical approval from the Weill Cornell Medicine Institutional Review Board and in accordance with all applicable regulations. Informed consent was obtained from the volunteer. The best performing coil array in terms of ease of manufacturing, thickness, and functional stability was determined to be the DIW coil (see results) and was used for in vivo imaging. Figure 6a shows a photograph of a standard commercial 8-channel knee coil array, which was used as a reference in this study. The weight of the coil is around 8 kg and it is fully rigid. In contrast, Figure 6b shows a photograph of the proposed stretchable dual-channel coil array positioned directly on the knee of a volunteer conforming to the anatomy. In Figure 6 the proposed coil is shown in its (b) relaxed and (c) unidirectionally stretched configurations, respectively.

The imaging was performed with the dual-channel coil array in the neutral (0%) and stretched (15%) positions. A fast spin echo (FSE) sequence with the following parameters was used: time to repeat *TR* = 4500 ms, echo time *TE* = 8.2 ms, field of view *FOV* = 18 cm, pixel size 0.4 × 0.6, echo train length *ETL* = 9, bandwidth *BW* = 83.3 kHz, number of averages *NEX* = 1, slice thickness = 1 mm. The SNR maps were calculated by following the same procedure as described in the previous section.

## 3. Results

### 3.1. Simulations

Figure 7 summarizes the simulation results for *S_ij_*-parameters and resonance frequency *f*_0_ with respect to coil stretching. From Figure 7a, we observe that the ideal overlap distance is 12 mm, the overlap value where the *S*_21_ parameter reaches its minimum of −13.5 dB. This overlap value was used in all further simulations, prototype fabrications, benchtop, and imaging tests. Figure 7b–d demonstrates simulation results on the *S_ij_*-parameters and resonance frequency *f*_0_ for coil array stretching from 0% to 50%. Figure 7b–c show the *S*_11_- and *S*_21_-parameter changes over a 20 MHz bandwidth with respect to coil stretching from 0% to 50% with a 10% increment. As shown in Figure 7d, when measured at the resonance frequency of 128 MHz, the *S*_11_-parameter stays below −10 dB for all degrees of stretch that were tested, and the *S*_21_-parameter stays below −10 dB for degrees of stretch up to 40%, indicating good coil element isolation. Figure 7e shows the change in resonance frequency *f*_0_ when stretched. We observe that the frequency first shifts upward for smaller degrees of stretch (<20%), followed by a downward shift for larger degrees of stretch (>20%). The maximum frequency variation is calculated to be 1.3%, from 127.7 MHz to 129.4 MHz. Figure 7f shows that the coil-to-coil distance does not affect the coupling between the coil elements (indicated by the relatively stable *S*_21_ parameter). Although the coil matching (indicated by the *S*_11_-parameter) does change with coil-to-coil distance, it stays below −15 dB for all the values studied.

Figure 8 shows simulated coil element sensitivity profiles (*B*_1_^−^-maps) of individual coil element (a)–(f) and the dual-channel coil array (g)–(i) when subjected to various stretching levels from 0% to 50%. The slight asymmetry in the field distribution is a well-known phenomenon attributed to polarization effects of the *B*_1_ field, which begin to emerge at higher *B_0_* field strengths (such as 3 T) and in more conductive samples [46].

### 3.2. Fabrication

#### 3.2.1. Single Layer Casting

Figure 9 shows photographs of the (a) 3D-printed mold and (b,c) fabricated coil elements with inserts zooming in on the overlap area. Figure 9b shows the fabricated coil array before the jumper wire was inserted, highlighting the absence of contact between the perpendicular conducting traces, with one continuous conductive trace (going from top left to bottom right) and one split trace (going from bottom left to top right). To establish the necessary electrical connection, a thin flexible silicone tube was used to bridge the gap in the loop element (the bottom left and top right channels in the inset (b)), thus creating a “jumper wire”. To secure the silicone tube in place and prevent any liquid metal from spilling, we utilized silicone epoxy (Sil-Poxy^TM^), as shown in the inset of Figure 9c. In total, two such jumper wires were implemented for this dual-channel array (Figure 9c).

#### 3.2.2. Double Layer Casting

Figure 10a shows the photograph of the 3D printed mold of the DLC coil. Figure 10b,c shows two fabricated DLC coil elements and Figure 10d shows the resulting DLC dual-channel coil array, where the two single-element loops are overlayed to achieve critical overlap decoupling.

#### 3.2.3. Direct Ink Writing

Figure 11a shows a photograph of the DIW printer while printing the channel walls. The fabricated DIW coil elements are shown in Figure 11 when they are positioned next to each other (b) and when they are critically overlapped (c). The advantage of DIW is the markedly reduced total thickness of the coil array (0.6 mm for DIW compared to 3 mm and 6 mm for the SLC and DLC methods, respectively). Moreover, from the fabrication point of view, the DIW technique uses less steps and thus is much faster and produces more repeatable results.

While all tested fabrication techniques produced functional prototypes, significant differences were observed in both the process and outcome for each method. Firstly, the SLC prototype exhibited electrical instability due to the added, mechanically vulnerable, jumper wire. Although the silicone tubes used were flexible, they lacked stretchability, leading to compromised electrical connections during extensive stretching. Secondly, the DLC prototype was twice as thick as the SLC (6 mm for the DLC compared to 3 mm for the SLC) and required double the time and polymer material for fabrication. It was also more difficult to stretch the DLC while maintaining correct overlap. Conversely, the DIW prototype exhibited a much thinner profile (only 0.6 mm) and demonstrated enhanced mechanical stability. The DIW technique used less material, required less fabrication time, eliminated the need for a 3D printed mold, and the overall automated fabrication process delivered more consistent results. Consequently, the DIW prototype emerged as the mechanically superior prototype and was used in in vitro and in vivo testing.

### 3.3. Benchtop Measurements

Figure 12 shows the measured frequency shift with respect to the degree of stretch for (a) SLC, (b) DLC, and (c) DIW coil arrays. In the SLC case (a), the two curves representing the two different channels do not align with each other, which is thought to be attributed due to the unevenness introduced by the jumper wire connectors. In the DLC case (b), the two curves align well with each other since the two elements are identical by design. The frequency variation in the SLC and DLC prototypes is higher than the shift seen in simulation, which is attributed to fabrication error/uncertainty. On the other hand, the DIW coil array (c) performs very closely to the simulated prediction. Figure 7e, with the frequency shifting upwards for small stretching levels (<12%) and shifting downwards for higher stretching levels. The maximum recorded frequency shift is from 127.6 MHz to 128.6 MHz for channel 1, and from 127.4 MHz to 128.2 MHz for channel 2, which corresponds to 0.8% of variation from the center frequency of 128 MHz.

### 3.4. In Vitro Imaging

Figure 13a–c compares SNR maps of the three coil arrays measured through the central axial slice. The SNR of all three coils is very similar with an average of 43 ± 4 inside a 1 cm ROI at the depth of 5 cm inside the phantom (as shown in Figure 8a). The DLC coil has the smallest SNR among all the methods, which is probably due to the increased coil thickness and associated signal losses. Figure 13d–f shows the signal measured along two vertical lines passing through the center of the coil elements (left and right white vertical lines) and along one horizontal line passing through the phantom at a depth of 1 cm (black horizontal line). These images demonstrate that all three techniques produce similar sensitivity maps and nearly identical SNR profiles. The DIW technique is favored due to its significantly easier fabrication as well as its reduced visibility in images—the coil array is much thinner (0.6 mm) and uses smaller amounts of polymer to produce. Anecdotally, we also propose to suppress the appearance of silicone polymer in MR images [47].

Figure 14 shows (a) simulated and (b) measured sensitivity maps of the DIW coil array under various degrees of stretch from 0% to 32% using increments of 1 cm. It is to be noted that the added stretch test setup in Figure 14 resulted in the coil array being located at 1 cm above the phantom, further from the phantom than in Figure 13. To match this experimental setup, the 3D numerical model was adapted accordingly, and the simulated sensitivity map was zoomed in to match the experimental FOV. The simulations (a) agree well with the experiments (b), and coil sensitivity is maintained throughout. Minor streaking artifacts can be observed on the measured SNR maps (b) above the phantom; however, they do not affect the image quality inside the phantom. The artifacts were eliminated in all other imaging experiments by changing the readout direction in the MRI sequence.

### 3.5. In Vivo Imaging

Figure 15 shows central (a)–(c) axial and (d)–(f) sagittal SNR maps of a healthy knee acquired with (a,d) a standard, rigid, commercial 3T T/R knee coil array by GE Healthcare; and with the proposed stretchable coil array in (b,e) unstretched and (c,f) stretched positions. In the axial slices, several representative regions of interest (ROIs) were selected as shown in Figure 15a, and the average SNR values within these ROIs were plotted in Figure 15g for comparison. We observe that the SNR of the proposed coil is more than tripled in ROI Top compared to the standard coil due to a more conformal anatomical fit. The SNR on the periphery (ROIs Left and Right) is also improved by up to 60%. This significant SNR improvement is attributed to the inherent conformal nature of the proposed stretchable coil array. Tightly fitting designs reduce the distance from the coil to the anatomy, which minimizes signal loss. Ultimately, better coil loading and optimal coverage of the ROIs are achieved. This prototype alone, with only two channels, surpasses the SNR in the ROI of the commercial coil with 8 channels significantly. While this work focuses on the fabrication aspects of a stretchable array, demonstrating the most suited and best performing modality to construct these coils, we expect even higher SNR gains when moving to a more practical array with a larger channel count in our upcoming research. In the sagittal slice, the proposed coil array produced up to 50% higher SNR in an ROI within the Hoffa’s pad with the SNR improving from 40 (commercial coil) to 52 (unstretched) and 60 (stretched). Even though the comparison between the two coil arrays (standard 8-channel and proposed 2-channel) may be slanted in favor of the commercial coil due to its higher intrinsic SNR, we achieve notably improved SNR due to the proximity to the anatomy of the proposed coil concept.

## 4. Discussion

Our simulations and experimental validation tests indicate the feasibility of constructing a stretchable self-tuning coil array based on our previously proposed coil design [32]. Coil prototypes produced using the SLC fabrication technique suffered from connectivity issues at the coil overlap area, as the interface between the liquid metal traces and the jumper wires was not stable at larger stretching levels (>20%). To mitigate this issue, we attempted other connection configuration using a copper wire instead of a liquid metal-filled silicone tube; however, this still did not provide sufficient stability under stretching. Coil prototypes produced using the DLC fabrication technique were too thick to handle (~6 mm) which led to a somewhat decreased ease of stretchability as well as potential SNR loss due to the increased background noise from the added material. Moreover, the DLC technique required double the time and material quantity compared to the SLC method. The coil prototypes produced using the DIW technique were more mechanically robust, required fewer fabrication steps, and were significantly thinner (0.6 mm) compared to the other prototypes while also resulted in the reduction in the unwanted signal from the polymer substrate. The subsequent systematic benchtop analysis of the three different fabrication methods indicates that all three fabrication techniques produce similarly performing coil arrays in terms of the resonance frequency and *S*-parameters. Our experimental *S*-parameter and coil sensitivity measurements agree well with the simulations.

It is to be noted that the commercial knee coil we have access to and use here consists of 8 elements compared to the two in our stretchable prototype, which slants the comparison in favor of the commercial coil because of its intrinsically higher SNR. Yet, we observe significantly improved performance with our coil concept, most notably due to its built-in proximity to the anatomy. This advantage in sensitivity is expected to increase even further with a larger number of elements. The goal for this paper was to establish optimized fabrication methods while still showcasing performance in vivo. In the future, a performance comparison with coils of similar caliber will be carried out with the full stretchable multi-channel array in hand.

The DIW coil array can still be visible in MR images, as can be seen in, e.g., Figure 15. As mentioned in the literature [48,49,50], RF coil parts often contain proton-rich engineering plastics and may contribute to a significant background noise when trying to measure short-T_2_ signals. Thus, in its current implementation, the proposed stretchable coil array may suffer from potentially high background noise and/or artefacts when ultra-short TE (uTE) sequences are used. However, the added value of softness and stretchability might outweigh the limitations, enable typically unavailable applications, such as dynamic imaging, and improve patient comfort. Moreover, other fabrication techniques and polymer recipe optimizations may lead to reduced polymer signal and can potentially enhance mechanical and electrical properties [51].

With the significant reduction in coil thickness from 3 mm to 0.6 mm used here, one might ponder whether it is conceivable to further push the limit of coil thickness using the DIW technique and achieve an even thinner coil. To this goal, we would like to make a few comments: first, the resistance *R* of a planar trace of liquid metal depends on the width *w*, height *h*, and length *l* of the metal trace as follows R=(ρ·l)/(w·h), where ρ is the resistivity of the metal. Second, skin depth plays a major role at ultra-thin layer thicknesses, increasing resistivity. The calculated skin depth of Ga liquid metal at the frequency of 128 MHz is ≈23 µm. Higher conductor resistance results in lower coil performance. Thus, the coil dimensions must be selected such that to maintain low coil resistance. In [42], the effect of the liquid metal trace height (conductor thickness) on the resonance frequency and quality factor *Q* of a coil was studied. It was found that although the resonance frequency was not affected by the thickness of the liquid metal, the quality factor *Q* was decreased in coils with liquid metal trace widths below 500 µm. This conclusion supported our selection of the 0.5 mm liquid metal channel height and limited the total thickness of stretchable coil.

Regarding the simulation model used, the parametrization with the constraint $\lambda_{x}\cdot \lambda_{y}\cdot \lambda_{z}=1$ yields a good approximation for uniaxial stretching. However, for stretching along more than one axis or to include effects not covered by this approximation, the coil can be simulated using a combination of the electromagnetic and solid mechanics modules in COMSOL. In this proof-of-concept work, we focus only on a unidirectional stretching in *x*-direction.

In this work, we studied a dual-channel coil array and explored several fabrication techniques that allow for overlapping no-contact conductors. This demonstrates the feasibility of scaling our previously proposed single-channel coil element [32] to a multi-channel coil array. With the focus of this work being on the optimization of the coil fabrication methods, we demonstrated in vivo coil performance only in one volunteer. Future work will involve the fabrication of a clinically practical array with higher channel count along with a dedicated study population. We will use this array for comparative studies between different age and sex groups, as well as to assess patient comfort levels when compared to the standard of care. Moreover, future work includes applying this coil concept to other anatomies such as long bone, wrist, neck, and shoulder imaging.

## 5. Conclusions

In this work, we systematically analyzed the feasibility of a multi-element self-tuning stretchable coil array. We investigated and compared three different fabrication methods and found that the DIW technique enables the reliable production of very thin (0.6 mm) coil arrays. All coil prototypes were tested on the bench and a frequency stability of 128 ± 0.6 MHz was recorded. All coil prototypes were tested in vitro in terms of their sensitivity maps and produced similar results (SNR = 43 ± 4), proving the feasibility and agreement between all three fabrication techniques. The best performing DIW coil array was used for in vivo knee imaging and outperformed a commercial knee coil in terms of its sensitivity and improved SNR by up to 50% inside the region of interest due to a tighter anatomical fit.

## Figures and Tables

**Figure 1 sensors-23-07588-f001:**
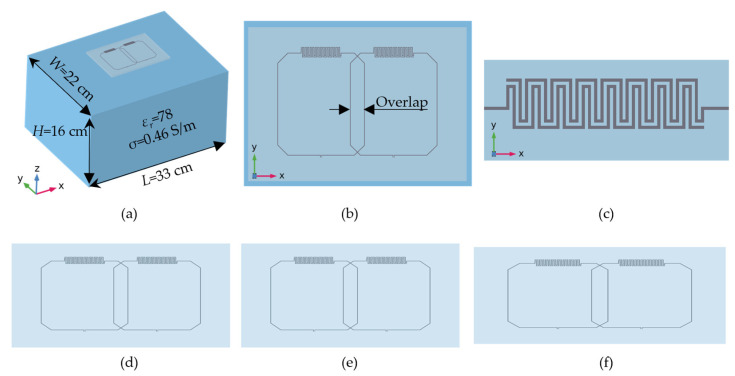
(**a**) Three-dimensional view of the simulation model. (**b**) Top view of the double-channel coil. (**c**) Enlarged view of the interdigital capacitor. Representative stretching configurations of (**d**) 10%, (**e**) 30%, and (**f**) 50% stretch.

**Figure 2 sensors-23-07588-f002:**
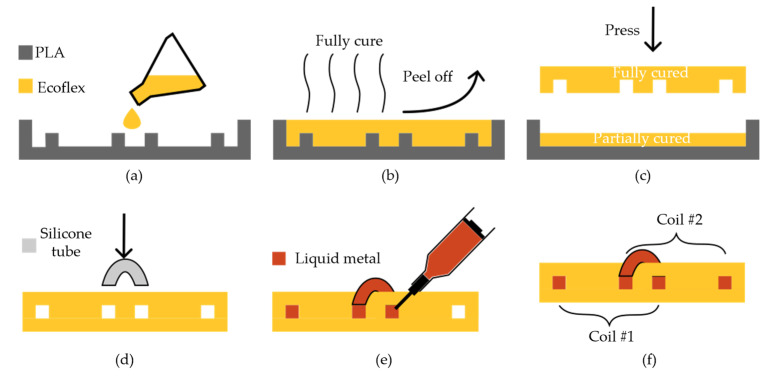
A step-by-step schematic of the SLC fabrication method. (**a**) Polymer casting into a prepared mold. (**b**) Curing and peeling the polymer off. (**c**) Bonding of top and bottom layers. (**d**) Silicone tube insertion. (**e**) Liquid metal injection. (**f**) Resulting SLC coil array.

**Figure 3 sensors-23-07588-f003:**
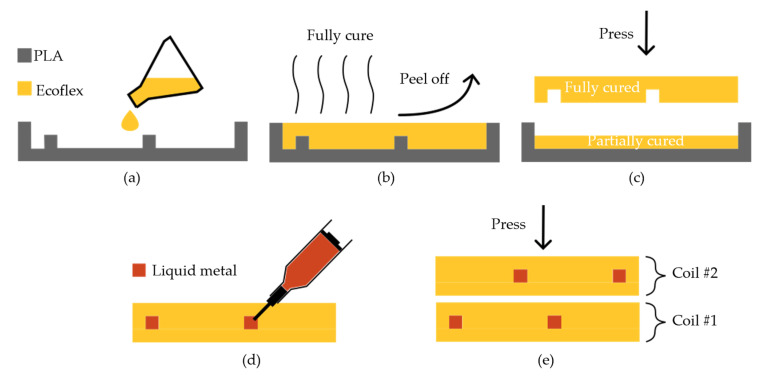
A step-by-step schematic of the DLC fabrication method. (**a**) Polymer casting into a prepared mold. (**b**) Curing and peeling the polymer off. (**c**) Bonding of top and bottom layers. (**d**) Liquid metal injection. (**e**) Resulting DLC coil array.

**Figure 4 sensors-23-07588-f004:**
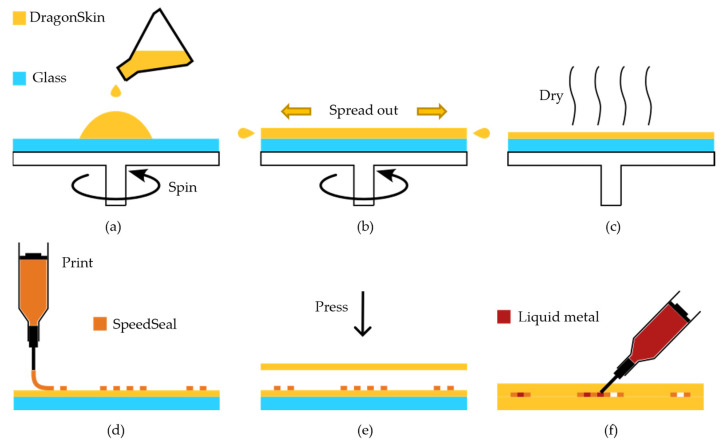
A step-by-step schematic of the DIW fabrication method. (**a**,**b**) Polymer spin coating. (**c**) Polymer curing. (**d**) Printing of the channel walls. (**e**) Bonding of the top and bottom layers. (**f**) Liquid metal injection.

**Figure 5 sensors-23-07588-f005:**
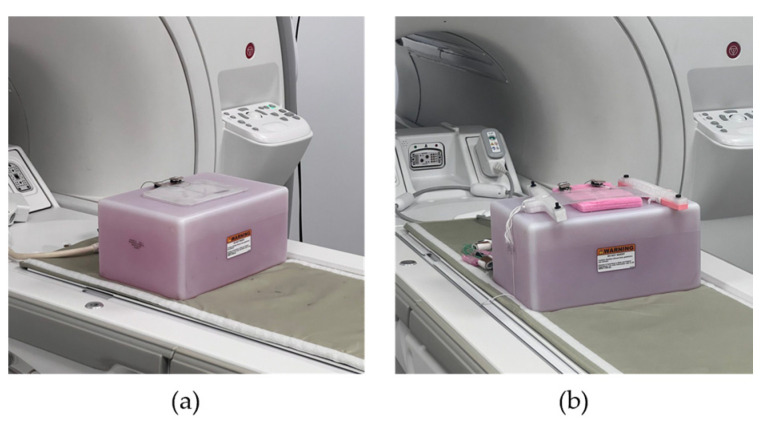
(**a**) DLC array on the phantom. (**b**) DIW array with a stretching rig on the phantom.

**Figure 6 sensors-23-07588-f006:**
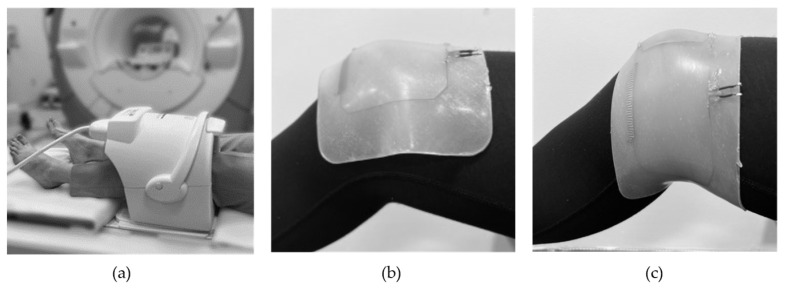
Photographs showing (**a**) a standard commercial knee coil and proposed stretchable coil in (**b**) relaxed and (**c**) stretched configurations.

**Figure 7 sensors-23-07588-f007:**
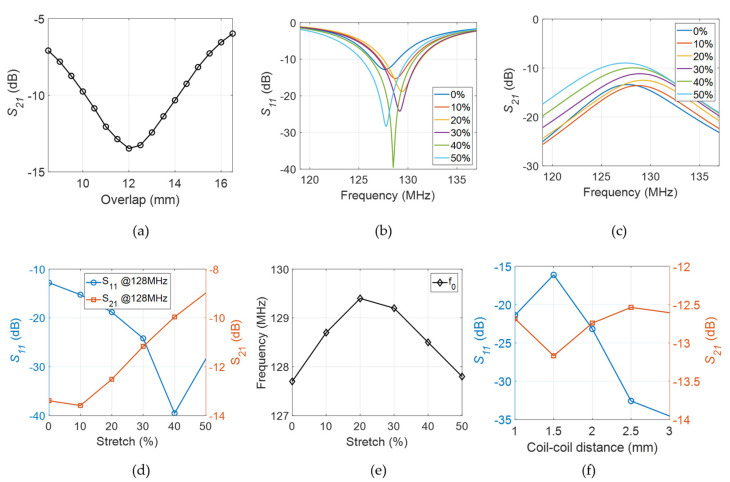
Simulation results of coil array stretching. (**a**) S_21_ parameter versus the overlap distance. (**b**) *S*_11_- and (**c**) *S*_21_-parameter changes with respect to coil stretching from 0% to 50% over a 20 MHz bandwidth. (**d**) *S*_11_- and *S*_21_-parameter changes with respect to stretching at the resonance frequency of 128 MHz (**e**) Resonance frequency *f*_0_ changes with stretching. (**f**) *S*_11_ and *S*_21_ parameters change with respect to coil-to-coil distance.

**Figure 8 sensors-23-07588-f008:**
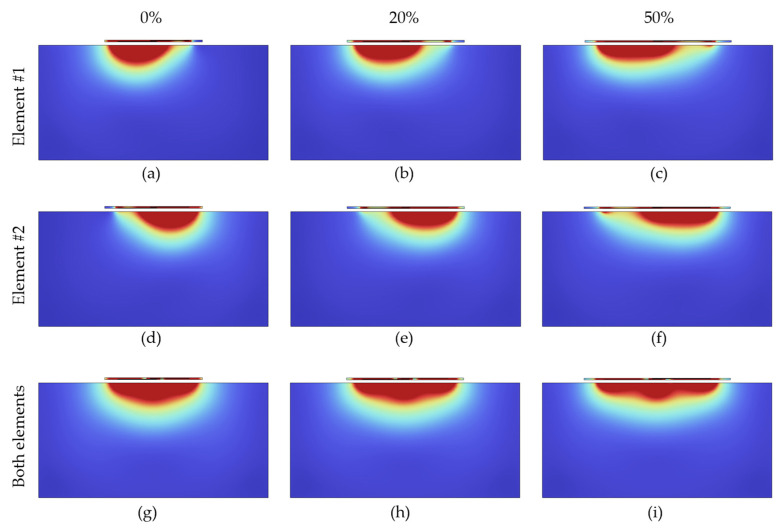
Simulated coil sensitivity profiles with stretching for individual coil elements (**a**–**c**) element #1 is ON, (**d**–**f**) element #2 is ON, and (**g**–**i**) for both elements ON (coil array).

**Figure 9 sensors-23-07588-f009:**
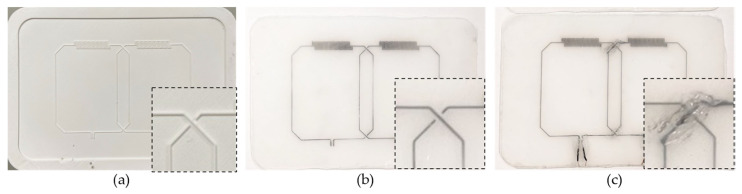
Photographs of the single layer casting (SLC) fabrication steps. (**a**) Three-dimensional printed mold, and fabricated coil array (**b**) before and (**c**) after jumper wires were inserted.

**Figure 10 sensors-23-07588-f010:**
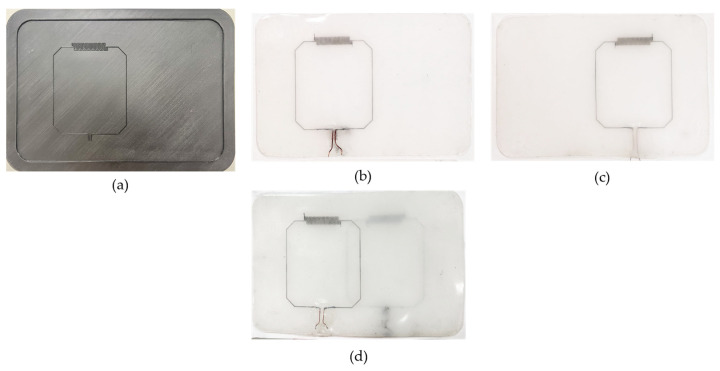
Double layer casting (DLC) fabrication method. (**a**) Three-dimensional-printed mold and fabricated (**b**,**c**) coil elements, and (**d**) coil array.

**Figure 11 sensors-23-07588-f011:**
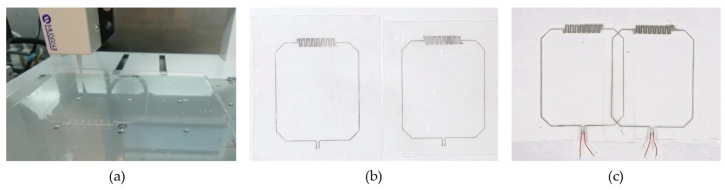
Direct-ink-writing (DIW) fabrication method. (**a**) DIW printer process and fabricated (**b**) coil elements and (**c**) coil array.

**Figure 12 sensors-23-07588-f012:**
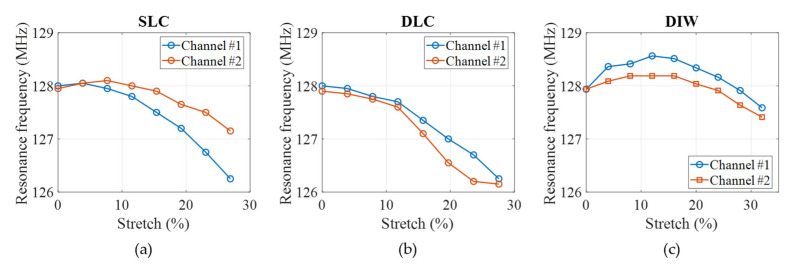
Measured resonance frequency *f*_0_ shift with stretch for (**a**) SLC, (**b**) DLC, and (**c**) DIW coil arrays.

**Figure 13 sensors-23-07588-f013:**
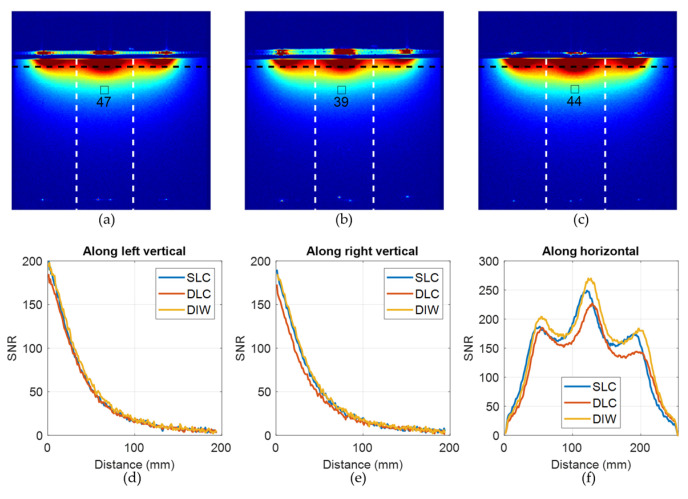
In vitro imaging using three different stretchable dual-channel coil prototypes. Central axial SNR maps of (**a**) SLC, (**b**) DLC, (**c**) DIW coil arrays. SNR versus distance into the phantom as measured along (**d**) left vertical and (**e**) right vertical lines going through the centers of the left and right coils, respectively; and (**f**) horizontal line (1 cm inside the phantom).

**Figure 14 sensors-23-07588-f014:**
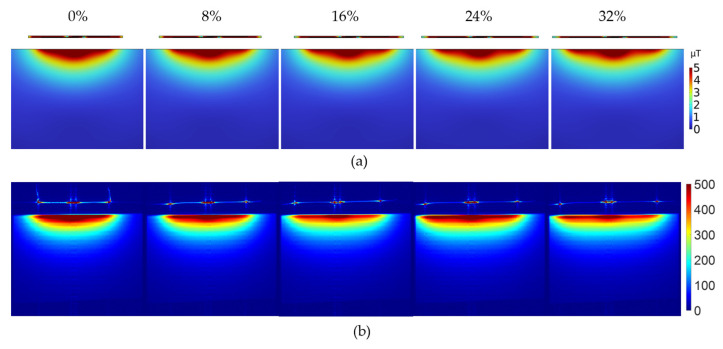
(**a**) Simulated and (**b**) measured sensitivity maps during stretching from 0% to 32%.

**Figure 15 sensors-23-07588-f015:**
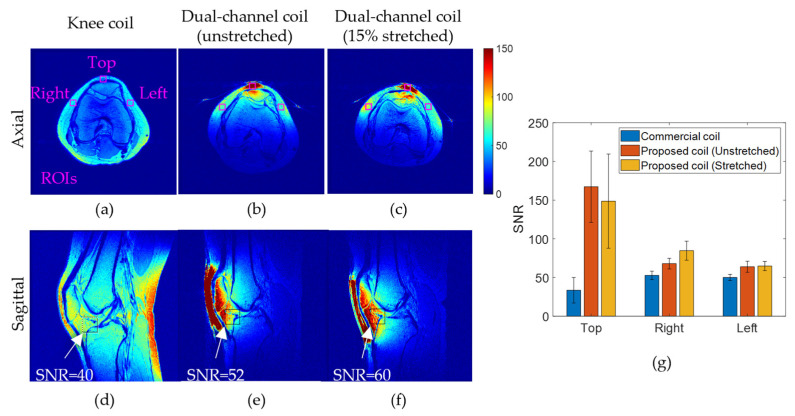
In vivo knee (**a**–**c**) axial and (**d**–**f**) sagittal SNR maps acquired with (**a**,**d**) a dedicated knee coil and with the dual-channel coil prototype in (**b**,**e**) unstretched and (**c**,**f**) stretched configurations. SNR within the Hoffa’s fat pad area is indicated with arrows on sagittal slices (**d**–**f**). (**g**) SNR comparison of different coil configurations in the ROIs. ROIs are shown in pink in (**a**).

## Data Availability

The data presented in this study are available upon reasonable request from the corresponding author.

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
