# Peer review of "Dual-Channel Stretchable, Self-Tuning, Liquid Metal Coils and Their Fabrication Techniques"

_sensors, 2023, doi:10.3390/s23177588_

Round 1

Reviewer 1 Report

This paper presents three fabrication techniques to manufacture dual-channel receive coil arrays. DIW technique enables the reliable production of very thin (0.6 mm) coil arrays, and it was used for in vivo knee imaging. Some questions are listed as below

1.      In section 2.1, some simulation results are suggested to be added. In section 3, some comparative analysis between simulation results and experimental results are suggested.

2.      An enlarged view of the serpentine coil, and some schematic images to realize the fabrication of different layers are suggested.

3.      What’s the influences of the thickness of the overlap to the coil performance? As the author said that DIW technique that enables the reliable production of very thin (0.6 mm) coil arrays have best performance. How to further reduce the thickness?

4.      In figure 11, the presented coil has much better performance, please give some discussions from the coil design and fabrication to explain the improvement.

good

Author Response

We thank the reviewer for their valuable comments and suggestions. We provide the responses to the comments below. All the changes are highlighted red in the updated manuscript. The changes attributed to particular reviewer’s comments are numbered and flagged as well.

This paper presents three fabrication techniques to manufacture dual-channel receive coil arrays. DIW technique enables the reliable production of very thin (0.6 mm) coil arrays, and it was used for in vivo knee imaging. Some questions are listed as below

R1.1      In section 2.1, some simulation results are suggested to be added. In section 3, some comparative analysis between simulation results and experimental results are suggested.

Thank you for your suggestions. We updated Figure 1 to give better detail on the simulation model and added corresponding descriptions to the main text. As suggested, we updated the results to highlight the comparative analysis. In Figure 12, we compare the resonance shift variation between prototypes where we observe that the resonance shifts the least with the thinnest coil (the DIW prototype). Moreover, this measured frequency behavior is in agreement with the simulations (Figure 7(e)). Additionally, in Figure 14 we compare the simulated and measured coil sensitivities with varied stretching. We also added an analysis studying the influence of layer thickness on coil decoupling (see R2.3), with the result that decoupling is maintained in all three prototypes, no matter the thickness of coil elements (Figure 7f).

R1.2      An enlarged view of the serpentine coil, and some schematic images to realize the fabrication of different layers are suggested.

Thank you. We included an enlarged view of the interdigital capacitor design for better model illustration (Figure 1c). We also created new schematic images of each fabrication process (Figures 2,3,4) and updated the corresponding text to clarify the individual steps of the fabrication process.

R1.3      What’s the influences of the thickness of the overlap to the coil performance? As the author said that DIW technique that enables the reliable production of very thin (0.6 mm) coil arrays have best performance. How to further reduce the thickness?

This is an interesting question. We have included additional simulation results (Figure 7(f)) to investigate whether “vertical” coil-to-coil distance (i.e., layer thickness) affects overlap decoupling and resonance stability. From our experiments, we observe that the resonance frequency remains most stable with the thinnest coil (Figure 12). We also see that thinner coils, while performing best otherwise, do not offer any specific advantages in terms of decoupling performance. Nevertheless, we added a paragraph in the Discussion section to address practical limitations in coil thickness.

R1.4.      In figure 11, the presented coil has much better performance, please give some discussions from the coil design and fabrication to explain the improvement.

Thank you for the suggestion. We now explain that signal to noise ratio is greatly improved when the distance from the coil to the anatomy is minimized because signal loss is minimized. In fact, we surpass the performance of an 8-channel commercial coil with 2-channels alone, simply because we are able to get close to the surface of the knee. We expect even greater SNR gains with our final, more practical, multi-channel array. This paper studied fabrication methods first and foremost, a key step in the road to this final, clinically implemented, array.

Reviewer 2 Report

In this manuscript, the authors reported dual-channel flexible and stretchable radiofrequency coils with three fabrication techniques and demonstrated bench testing, in vitro, and in vivo imaging. The results of the manuscript are quite interesting. I recommend the publication of this paper in Biomedical sensors once the following points are addressed:

(1) For the liquid metal, is it gallium indium (GaIn)? Can the author mention this somewhere in the experimentation section? It can be in the 2.2 fabrication technique or anywhere the author think is more suitable earlier rather than line 171 (2.2.4).

(2) In Figure 1, the author mentioned designed with and without the jumper wire. Can the author tell what is the difference between these two and what is the advantage of having a jumper wire? Is it the one without the jumper wire means no liquid metal?

(3) in the section 2.2.3 direct ink writing, the author mentioned the use of DragonSkin TM 30 silicon as a base. Can the author justified the use of this ecoflex instead of other type of ecoflex such as 00-30 or 00-50? Is it because it is easier for the adhesion for the direct writing or because of its stretchability?

(4) How many volunteer that author used in the experiment involved relaxed and stretched condition of the knee coil? Only one? Is it enough? Can you justify? 

Author Response

We thank the reviewer for their valuable comments and suggestions. We provide the responses to the comments below. All the changes are highlighted red in the updated manuscript. The changes attributed to particular reviewer’s comments are numbered and flagged as well.

In this manuscript, the authors reported dual-channel flexible and stretchable radiofrequency coils with three fabrication techniques and demonstrated bench testing, in vitro, and in vivo imaging. The results of the manuscript are quite interesting. I recommend the publication of this paper in Biomedical sensors once the following points are addressed:

R2.1 For the liquid metal, is it gallium indium (GaIn)? Can the author mention this somewhere in the experimentation section? It can be in the 2.2 fabrication technique or anywhere the author think is more suitable earlier rather than line 171 (2.2.4).

Thank you for the suggestion. We used GaIn and introduced it earlier in the manuscript. We also added a few sentences on the reasons behind using GaIn metal for this work in Section 2.2.4.

R2.2 In Figure 1, the author mentioned designed with and without the jumper wire. Can the author tell what is the difference between these two and what is the advantage of having a jumper wire? Is it the one without the jumper wire means no liquid metal?

We apologize for the lack of clarity in the description of the jumper wire fabrication method. In Figure 9, we meant to show the coil layout before and after jumper wire was inserted. The jumper wire is an essential element in the SLC method as it connects otherwise electrically separate parts of the coil. We have updated Figures 2 and 9 and the text accordingly to remove the any confusion.

R2.3 in the section 2.2.3 direct ink writing, the author mentioned the use of DragonSkin TM 30 silicon as a base. Can the author justified the use of this ecoflex instead of other type of ecoflex such as 00-30 or 00-50? Is it because it is easier for the adhesion for the direct writing or because of its stretchability?

Thank you for pointing this out. Material selection is one of the most important aspects of the fabrication process here, and we have now included more information, clarification, and justification. The single- and double-layer coils were indeed fabricated with Ecoflex 00-30 because of its high stretchability, biocompatibility, and ease of fabrication. The thinner DIW coil required a material with higher tensile strength, and DragonSkin was found to be a suitable candidate. We included these justifications in the corresponding sections 2.2.1, 2.2.3, and 2.2.4.

R2.4 How many volunteer that author used in the experiment involved relaxed and stretched condition of the knee coil? Only one? Is it enough? Can you justify? 

Thank you for your comment. In this study, we focused on optimizing coil fabrication methods, and thus, for demonstration purposes, we only scanned one volunteer. However, in future work we plan to develop a coil with a larger channel count that is clinically practical. For this coil, a dedicated study population of diverse age and gender distribution will be recruited.